# Influence of Pre-Incubation of Inoculum with Biochar on Anaerobic Digestion Performance

**DOI:** 10.3390/ma16206655

**Published:** 2023-10-11

**Authors:** Marvin T. Valentin, Kacper Świechowski, Andrzej Białowiec

**Affiliations:** 1Department of Applied Bioeconomy, Wrocław University of Environmental and Life Sciences, 51-630 Wroclaw, Poland or marvin.valentin@bsu.edu.ph (M.T.V.); kacper.swiechowski@upwr.edu.pl (K.Ś.); 2Benguet State University, Km. 5, La Trinidad, Benguet 2601, Philippines

**Keywords:** biochar, glucose, pre-incubation, inoculum, methane

## Abstract

The application of biochar as an additive to enhance the anaerobic digestion (AD) of biomass has been extensively studied from various perspectives. This study reported, for the first time, the influence of biochar incubation in the inoculum on the anaerobic fermentation of glucose in a batch-type reactor over 20 days. Three groups of inoculum with the same characteristics were pre-mixed once with biochar for different durations: 21 days (D21), 10 days (D10), and 0 days (D0). The BC was mixed in the inoculum at a concentration of 8.0 g/L. The proportion of the inoculum and substrate was adjusted to an inoculum-to-substrate ratio of 2.0 based on the volatile solids. The results of the experiment revealed that D21 had the highest cumulative methane yield, of 348.98 mL, compared to 322.66, 290.05, and 25.15 mL obtained from D10, D0, and the control, respectively. Three models—modified Gompertz, first-order, and Autoregressive Integrated Moving Average (ARIMA)—were used to interpret the biomethane production. All models showed promising fitting of the cumulative biomethane production, as indicated by high $R^{2}$ and low RMSE values. Among these models, the ARIMA model exhibited the closest fit to the actual data. The biomethane production rate, derived from the modified Gompertz Model, increased as the incubation period increased, with D21 yielding the highest rate of 31.13 mL/gVS. This study suggests that the application of biochar in the anaerobic fermentation of glucose, particularly considering the short incubation period, holds significant potential for improving the overall performance of anaerobic digestion.

## 1. Introduction

Anaerobic digestion (AD) is currently a relevant and compelling topic owing to its various advantages and positive impacts on the environment. It continues to offer solutions in the field of waste management and climate change mitigation, as well as adhering to the circular economy approach and creating various economic opportunities. AD can be used to treat a vast array of substrates such as food waste (FW) [1]; the organic fraction of municipal solid waste (OFMSW) [2]; animal excretion such as cow manure [3] and chicken dung [4]; wastewater [5]; and various biomasses [6,7].

Biochar (BC) has emerged as a valuable supplement with the potential to significantly enhance methane production during the anaerobic digestion of various substrates [3,8,9]. Several findings asserted the positive impact of integrating biochar into the AD system. Li et al. (2022) observed a 30.0% improvement in the lag phase during the co-digestion of corn straw and sewage sludge supplemented with biochar at 5 g/kg [10]. Pa et al. (2022) reported an 81.3% increase, relative to the control, in methane production after the addition of cow manure biochar [3]. Moreover, the introduction of BC can also stimulate the growth of important microorganisms in the reactor [11], further enhancing the efficiency of the anaerobic digestion process.

The anaerobic digestion process relies on a consortium of diverse microorganisms, each playing a crucial role in the sequential stages of hydrolysis, acidogenesis, acetogenesis, and methanogenesis. In the study of Wang et al. (2022), wheat straw biochar fostered an environment conducive to microbial activity [8]. The biochar was able to host numerous cocci and microbiota, consequently enhancing the utilization rate of the biomass being treated [8]. This phenomenon of microbial enrichment on the surface of biochar has also been consistently attested across various studies [12,13]. Importantly, the attachment of microorganisms onto BC’s surface facilitates the formation of biofilm [14]. Several studies have identified specific microorganisms that thrive in the presence of BC. For instance, *Methanosarcina* [10,15,16] and *Methanosaeta* [13,16] are common microorganisms that are enriched when BC is introduced into the system. Luo et al. (2015) found that BC could establish a methanogenic zone on its pore spaces, which is the preferred location for *Methanosarcina* [17]. *Methanosarcina*, being acetoclastic and hydrogenotrophic [6], is a versatile and multi-nutrient (H_2_, CO_2_, acetic acid)-using genus for CH_4_ production [4,18]. Lee et al. (2016) found that attached biomass on carbon materials had more sequences within *Actinobacteria*, *Betaproteobacteria*, and *Deltaproteobacteria* [19]. Furthermore, the pore structure of BC can reduce ammonia inhibition and immobilize methanogens [3,17,20]. BC contains oxygen-containing functional groups (C−O, C=O, and C−OH) [21]. The presence of these functional groups enables BC to shuttle electrons in dual mode, as a donor and an acceptor, which eventually creates the phenomenon known as direct interspecies electron transfer (DIET) [22].

The addition of BC in AD accelerates methanogenesis by promoting the DIET phenomenon [1,3], which is associated with various characteristics, such as electrical conductivity (EC) [23], redox property [21], and surface oxygen functional groups [1]. In addition, BC supplementation can selectively enrich several bacteria and archaea favorable for the AD process [24]. Specifically, BC can serve as an alternative for C-type cytochromes and conductive pili to promote the electron transfer process [1].

Currently, no information reported in the literature has addressed the pre-mixing of biochar in the inoculum or digestate before the AD experiment. However, there are similar studies that examine the continuity of the biofilm formed in the BC during AD for use in long runs [25], which may provide insight into the importance of BC incubation before the AD process. The incubation period allows the BC to interact with the microbes present in the inoculum, but the optimum period must be considered as BC nutrient leaching may be weakened by long-term incubation [22]. The incubation of BC in the inoculum for a certain duration before its application in AD experiments can condition the BC to colonize microorganisms and facilitate biofilm formation. According to Johnravindar et al. (2020), biofilm formation allows the efficient degradation of organic matter and is associated with DIET occurrence [26]. Biofilm formation was observed from reactors supplemented with carbon materials [25,26,27]. He et al. (2020) observed stability in methane production, using the biochar biofilm as an inoculant, despite the removal of planktonic microorganisms in the reactor, and no extra inoculum or biochar was added [25]. Sunyoto et al. (2019) affirmed the role of biochar in enriching microbial growth and initiating biofilm formation [27]. Hence, in this study, it is speculated that the incubation of biochar in the inoculum would result in an enriched mixture favoring the anaerobic digestion of biomass. This could also lead to the possibility of recycling BC from previous AD experiments for subsequent use in the next AD process, especially since most studies have only considered the single use of biochar in AD [25].

This study was carried out to investigate the incubation of biochar in inoculum before the anaerobic fermentation of glucose. Three incubation periods were used: 21, 10, and 0 days. The performance of the anaerobic process was evaluated in terms of cumulative methane production, methane production rate, and pH variation.

A literature search performed using Web of Science revealed 822 articles that contained biogas and biochar in the keywords, from which recent review articles were identified [28,29]. To perform a keyword analysis, VOSviewer software was used [30]. The map was created based on bibliographic data with co-occurrence, all keywords, and full counting as the settings for the type of analysis and counting method. To ensure the relevance and significance of the keywords, the maximum number of occurrences was set at a threshold of 2. Within these keywords, “biochar” emerged as the most prominent keyword, exhibiting a robust link frequency of 165, closely followed by “biogas” with a link frequency of 131. The generated visualization map has a scale of 1.3 with the weights set to links. Additionally, the variation in label and line sizes was set to 1.0, with the maximum line strength capped at 1000. The resulting generated network map of the keywords that were most used in these studies is shown in Figure 1. The color and size indicate the type of cluster and the occurrence frequency of the individual keywords, respectively, and the line between the circles represents links [30]. The network map shows three major clusters. The first cluster (red) is the representation of the biochar, which is mostly related to BC sources. The second cluster (yellow) represents the common application of biochar. BC preparation is also grouped in the third cluster (blue). This information justifies the continuing evaluation of biochar application in biogas production, which substantiates the value of the current investigation.

## 2. Materials and Methods

### 2.1. Materials

The biochar used in this study was synthesized from sun-dried wheat straw obtained from the Swojec farm in Wrocław, Poland. The wheat straw underwent size reduction through the use of an electric stainless steel grinding machine. The ground wheat straw was pyrolyzed in a muffle furnace at a temperature of 900 °C for 60 min residence time, according to [13,31]. The digestate was acquired from the 1.0 MWel commercial agricultural biogas plant (Świdnica, Poland) for treating food waste and agricultural residues. The digestate underwent filtration to remove fibers and other solid materials such as plastics and stones [32]. The suspended liquid was set aside in a climatic chamber (Pollab, model 140/40, Wilkowice, Poland) at 4 °C and used as an inoculum in the anaerobic fermentation experiment. Glucose (substrate) was used as a carbon source in the experiment. The proportions of the inoculum and glucose were adjusted to an inoculum-to-substrate ratio (ISR) of 2.0 based on volatile solids (VS) [5,33,34,35,36,37,38]. Important properties of the inoculum, such as volatile solids, total solids, and pH, were determined.

### 2.2. Experimental Setup

The experimental setup followed the procedure previously conducted in the laboratory by Świechowski et al. (2022) [39]. An automatic methane potential test system (BPC Instruments AB, AMPTS^®^ II, Lund, Sweden) was used. The anaerobic fermentation was performed in 500 mL glass reactors with agitation [39,40] under mesophilic conditions (37 °C). The working volume of the reactor was 400 mL. The mixture comprised of 250 mL of inoculum, 3.0 g of glucose (ISR = 2), and 150 mL of nutrient solution. The 150 mL of nutrient solution contained (per liter) 0.2 g MgCl_2_.6H_2_O, 1 g NH_4_Cl, 0.1 g CaCl_2_, 0.2 g Na_2_S.9H_2_O, 2.77 g K_2_HPO_4_, 2.8 g KH_2_PO_4_, 0.1 g yeast extract, 5 mL trace element solution, and 2 mL vitamin solution [17,41]. The composition of the trace element solution (per liter) was 1000 mg Na2-EDTA.2H_2_O, 300 mg CoCl_4_, 200 mg MnCl_2_.4H_2_O, 200 mg FeSO_4_.7H_2_O, 200 mg ZnCl_2_, 80 mg AlCl_3_.6H_2_O, 60 mg NaWo_4_.2H_2_O, 40 mg CuCl_2_.2H_2_O, 40 mg NiSO_4_.6H_2_O, 20 mg H_2_SeO_4_, 200 mg HBO_3_, and 200 mg NaMoO_4_.2H_2_O [41]. The vitamin solution consisted of (per liter) 10 mg biotin, 50 mg pyridoxin HCl, 25 mg thiamine HCl, 25 mg D-calcium pantothenate, 10 mg folic acid, 25 mg riboflavin, 25 mg nicotinic acid, 25 mg P-aminobenzoic acid, and 0.5 mg vitamin B1 [41]. During the fermentation process, mixing occurred every hour for 3 min using the default mixing setting of the AMPTS to maintain homogeneity [40]. The produced biomethane was automatically measured and recorded by the AMPTS equipment.

Before the anaerobic fermentation, the initially prepared inoculum was mixed with biochar. This biochar-enhanced inoculum was subjected to three distinct incubation periods: 21 days, 10 days, and an immediate application (0 days). These experimental conditions were labeled as D21, D10, and D0, respectively, and C for the control. The control reactor did not receive biochar. The inoculum combined with biochar was carefully secured in a plastic container and was maintained at room temperature throughout the incubation period. The D0 treatment signifies the incorporation of biochar in the inoculum right at the commencement of the experiment, setting it apart from the other incubation periods.

### 2.3. Data Analysis

The biomethane generated from the glucose in the anaerobic fermentation was calculated using Equation (1).
(1)$$SMP(t)=\frac{{MP}_{M}-{{VS}_{DM}\times ASMP}_{D}}{S}$$
where SMP(t) is the specific methane production (mL/gVS) from the anaerobic fermentation of glucose at any time; ${MP}_{M}$ is the biomethane production from the mixture of inoculum, biochar, and glucose (mL); ${VS}_{D}$ is the volatile solid concentration of the inoculum in the mixture (gVS); ${ASMP}_{D}$ is the average specific biomethane production of the inoculum from the control reactor (mL/gVS); and S is the amount of VS of the glucose placed into the specific reactor (gVS).

The SMP was modeled using the Modified Gompertz [42,43,44] (Equation (2)) and first-order equation (Equation (3)). The variables in the models were estimated with the use of Python and were validated in Statistica v13.0 software (TIBCO Software Inc., Palo Alto, CA, USA). The Python code was prepared in the Jupyter Notebook.
(2)$$M\left(t\right)=P\times exp\left[-exp\left(\frac{R_{max}\times e}{P}x\left(\lambda -t\right)+1\right)\right]$$
where $M\left(t\right)$ is the cumulative biomethane production in mL/(gVS) at time, t; P  is the ultimate biomethane production in mL/(gVS); $R_{max}$ is the maximum biomethane production rate in mL/(gVS.day); λ is the duration of lag time (day); t is the processing time in day; and e is a mathematical constant (2.718282). In the first-order model, the maximum biomethane yield and the apparent hydrolysis degradation coefficient, $k_{d}$, were obtained (Equation (3)) [44]. The first-order kinetic model was used to characterize the hydrolytic process of the AD [2,45,46] at different conditions.
(3)$$B=B_{0}\;*\;(1-e^{-k_{d}t})$$
where B is the cumulative biomethane production at a given time t, $B_{0}$ is the ultimate biomethane potential yield, $k_{d}$ is the first-order disintegration rate, and t is the process time in days. Furthermore, a machine learning model, specifically the Autoregressive Moving Average (ARIMA) model, was used to analyze the trend in biomethane production. The data were divided into training data (70%), validation data (10%), and prediction data (20%). The accuracy of the models used in the validation and data prediction was assessed in terms of RMSE [47] and $R^{2}$ [48].

## 3. Results

### 3.1. Effects of BC on Methane Production

The cumulative biomethane production from the anaerobic fermentation of glucose, including the effects of biochar incubation in the inoculum, is shown in Figure 2. The production of biomethane started at day 0 in all reactors, except for the control groups, which highlights the fast response of the microbial community toward the feedstock [49]. Statistical analysis demonstrated that biomethane productions were statistically different (*p* < 0.05) among the treatments. Specifically, D21 had the highest biomethane production throughout the experiment. When the reactors were applied with D10 and D0, the biomethane yields followed a similar path until day 7. However, after day 7, D0 lags behind D10. The cumulative biomethane production for D21 was 348.98 mL, followed by 322.66 mL and 290.05 mL for D10 and D0, respectively. The trend shows an increase in biomethane production when increasing the incubation period of the biochar in the inoculum before the anaerobic fermentation. The observed differences in the biomethane production demonstrate the positive impact of incubating the biochar in the inoculum for different durations. The positive observation on the influence of biochar relative to the control is highly consistent with previous works reported elsewhere [3,10,50,51,52].

Zang et al. (2021) concluded that BC addition in their experiment did not cause an obvious leap in CH_4_ yield [15]. In this study, a significant difference in the produced biomethane was observed as a result of the incubation of biochar in the inoculum. The reactors (D0) that received biochar at the start of the experiment (without incubation) deviated significantly, particularly from day 7 until the end of the fermentation period. Furthermore, the biomethane yield from D21 was consistently higher throughout the experiment. This indicates that the biochar incubation over a longer period (21 days) in the inoculum could have established adaptation and perhaps developed a higher microbial community.

The results of the modified Gompertz model as applied to the biomethane production of glucose are shown in Table 1. The maximum biomethane production and production rate were highest in D21, with values of 355.20 mL and 31.13 mL/gVS, respectively. This indicates a higher potential for biomethane generation compared to the other treatments. All treatments exhibit a negative lag phase, suggesting a temporary decline in biomethane production before the exponential growth phase is entered. Lauzurique et al. (2022) obtained a negative lag phase with the use of the same model and found out that it had no physical meaning to the fit; rather, it is just a result of the mathematical structure of the model [5]. D21 has the highest RMSE, implying a higher degree of error in predicting biomethane production for this treatment. Furthermore, all treatments show a high $R^{2}$ value, which is within the acceptable range (0.75–1.0) [4], suggesting a good fit for the modified Gompertz model.

The first-order model provides information about the maximum biogas produced (*B_o_*) and the biodegradability constant (*k*) [32] for each treatment (Table 2). Similar to the modified Gompertz result, D21 has the highest biomethane production, of 372.96 mL, indicating a faster rate of biomethane generation compared to the other treatments. The predicted cumulative biomethane yield was higher than the actual value. In terms of error, D21 has the highest RMSE, suggesting a larger deviation between the model predictions and the actual data. The $R^{2}$ values indicate the good fit of the first-order model to the data, as they are high for all treatments.

The ARIMA model, in addition to the modified Gompertz and first-order models, was applied to further analyzed the trend of the biomethane production. The evaluation metrics (RMSE and $R^{2}$) of the model are given in Table 3. The optimum parameters for the ARIMA order (*p*, *d*, *q*) are also provided. The corresponding statistical parameters for this optimal condition in terms of RMSE indicate that D10, among the BC-supplemented reactors, has the lowest prediction error and deviation between the actual and predicted values. In terms of the proportion of variance in the data explained by the ARIMA model, all treatments have high $R^{2}$ values, suggesting the good fit of the ARIMA model.

All three models provided insights into the biomethane production process and exhibited reasonably good fitting to the data under various treatments. The modified Gompertz and first-order models offer more detailed parameter estimation, such as maximum biomethane production, maximum methane production rate, lag phase, and biodegradability. These parameters helped understand the dynamics of the biomethane production process. In terms of RMSE, all three models show varying levels of prediction errors across the treatments, with D21 having the highest error in all models. The $R^{2}$ values are consistently high for all treatments in all models, indicating a good fit and a high proportion of explained variance. The modified Gompertz and first-order models provide specific insights into maximum biogas production rate and biodegradability, while the ARIMA model assesses stationarity and provides statistical evaluation. Furthermore, in terms of the maximum production at D21, the modified Gompertz (355 mL/gVS) and the first-order (372 mL/gVS) models are reasonable owing to the theoretical biomethane production of glucose, which should be around 377 mL/gVS from the Buswell and Mueller stoichiometric formulas (Equations (4) and (5)).
(4)$$C_{c}H_{h}O_{o}H_{n}S_{s}+\left(c-\frac{h}{4}-\frac{o}{2}+\frac{3n}{4}+\frac{s}{2}\right)H_{2}O\overset{}{\to }\left(\frac{c}{2}+\frac{h}{8}-\frac{o}{4}-\frac{3n}{8}-\frac{s}{4}\right)CH_{4}\;+\left(\frac{c}{2}-\frac{h}{8}+\frac{o}{4}+\frac{3n}{8}+\frac{s}{4}\right)CO_{2}+nNH_{3}+sH_{2}S$$
(5)$$C_{c}H_{h}O_{o}+\left(c-\frac{h}{4}-\frac{o}{2}\right)H_{2}O\overset{}{\to }\left(\frac{c}{2}-\frac{h}{8}+\frac{o}{4}\right)CO_{2}+\left(\frac{c}{2}+\frac{h}{8}-\frac{o}{4}\right)CH_{4}$$
where $C_{c}H_{h}O_{o}H_{n}S_{s}$ is the elemental composition of the biomass that is comprised of carbon (C), hydrogen (H), oxygen (O), nitrogen (N), and sulfur (S); c, h, o, n, s are the percentage share of the volatile solids of biomass. Hence, with the complete degradation of 1.0 g VS of glucose, a theoretical quantity of 746.6 mL/g-VS biogas can be produced, with the production of the biomethane at the level of 377.95 mL/g-VS.

Further comparisons between the predicted and actual biogas production from the modified Gompertz, first-order, and ARIMA models obtained at D21 and D10 are shown in Figure 3 and Figure 4, respectively. As depicted in the figures, each model has different fittings to the actual data, especially at the beginning of the experiment. In the modified Gompertz, the predicted initial biomethane production at D21 was 44.83 mL/g-VS and 30.29 mL/g-VS at D10, which is far from the actual value of zero. The ARIMA and the first-order models had the same predicted values as that of the actual value (0 mL/g-VS) on day 0. The modified Gompertz showed a proportional increment during the first 8 days, giving a straight line of fit while the ARIMA model almost followed the actual data. The first-order model starts at zero but lags behind the actual data, particularly in the first 3 days. At the end of the anaerobic fermentation process (day 20), the predicted cumulative biomethane production from D21 was 348.98, and 349.02 mL/VS for the modified Gompertz and ARIMA models, respectively, which is closer to the measured value of 348.98 mL/gVS. However, the predicted higher value was at 354.39 mL/gVS (1.54%). At D10, the modified Gompertz and ARIMA predictions were likewise closer to the actual value (322.67 mL/gVS) while the first-order model predicted a higher value of 332.61 mL/gVS (3.08%). From this, considering the final cumulative biomethane production, the modified Gompertz model was more appropriate in this study compared to the first-order model. However, the modified Gompertz implied that biomethane was generated at day 0, when the value should have been zero.

### 3.2. Biomethane Production Rate

The daily biomethane production of the anaerobic fermentation of glucose under various conditions is shown in Figure 5. The impact of the addition of biochar is evident on day 1, where the peak biomethane production rate was observed from all the reactors with D21 being the highest, with 83.33 mL/g-VS/day, followed by D10 with 65.71 mL/g-VS/day. A gradual decrease in the biomethane production rate was observed until day 4 and a slight rise attempting to approach a second peak was observed at day 5 for all reactors. A similar trend was reported by Li et al. (2022), except that peak production started at day 6 [10]. The high biodegradability in D21 can be due to the high microbial population developed during the incubation period. He et al. (2020) observed that microorganisms and functional methanogens were enriched and proliferated in recycled biochar [25]. This could suggest that, in the present study, the biochar that was inoculated and digested for a longer period had been enriched with functional microorganisms that resulted in higher biomethane production.

### 3.3. System Stability

The behavior of the anaerobic fermentation of glucose in terms of stability, as indicated by pH variation induced by the different biochar incubation periods, is shown in Figure 6. The reactors that received biochar showed lower pH throughout the anaerobic fermentation compared to the non-amended reactors. The initial pH was 7.61, 7.59, 7.70, and 8.15 for D21, D10, D0, and C, respectively. During the first 10 days, the BC-amended reactors showed a gradual shift to alkalinity conditions and were stable until day 20. Conversely, the pH in the control reactor continued to increase, indicating that the control becomes more alkaline during the process time. This indicates that biochar was effective in promoting stability in the reactor, which confirms the findings of Lu et al. (2016) [13]. Previous reports have consistently demonstrated that the supplementation of BC effectively maintained an optimum pH level while non-BC reactors were acidic [13,53]. This current study aligns with these findings, showcasing the positive effect of BC, except that the control was alkaline. Pan et al. (2019) reported a similar observation where non-amended reactors showed alkaline pH levels [53].

## 4. Conclusions

This study explored the potential advantages of incubating the biochar into the inoculum before subjecting to anaerobic fermentation. The results of the investigation revealed that the D21 treatment yielded the highest performance, with a total biomethane volume of 348.98 mL accumulated at the end of the experiment. The stability of the system, as indicated by the pH, demonstrated that the reactors with biochar maintained a pH range of 7.6 to 8.3 compared with the control, which reached a pH of 8.6. The biomethane production was modeled using modified Gompertz, first-order, and ARIMA models. The three models showed strong fitting of the data, as indicated by high values of R^2^ and low RMSE values. However, since biochar incubation showed better performance than when mixed during the experiment, a future study is recommended in which the incubated BC is subjected to microbial analysis.

## Figures and Tables

**Figure 1 materials-16-06655-f001:**
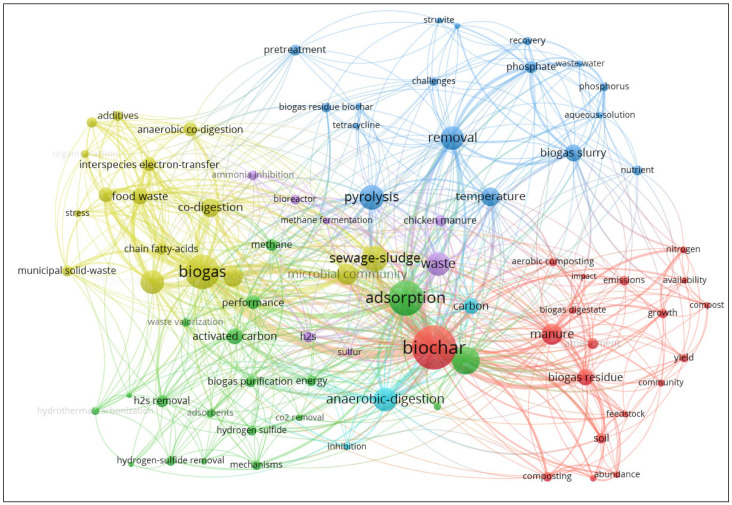
The network map of keyword co-occurrence generated from VOSviewer software v1.6.19.

**Figure 2 materials-16-06655-f002:**
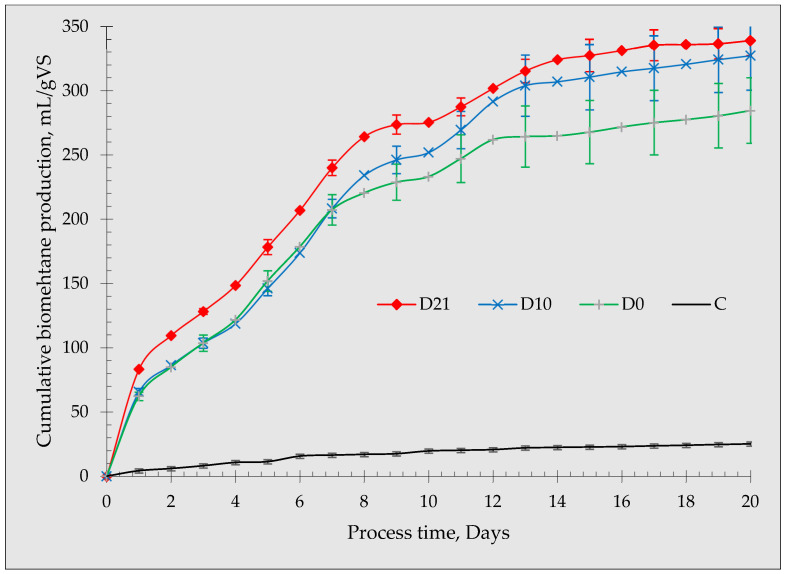
Cumulative biomethane production from the anaerobic fermentation of glucose supplemented with biochar with different incubation periods.

**Figure 3 materials-16-06655-f003:**
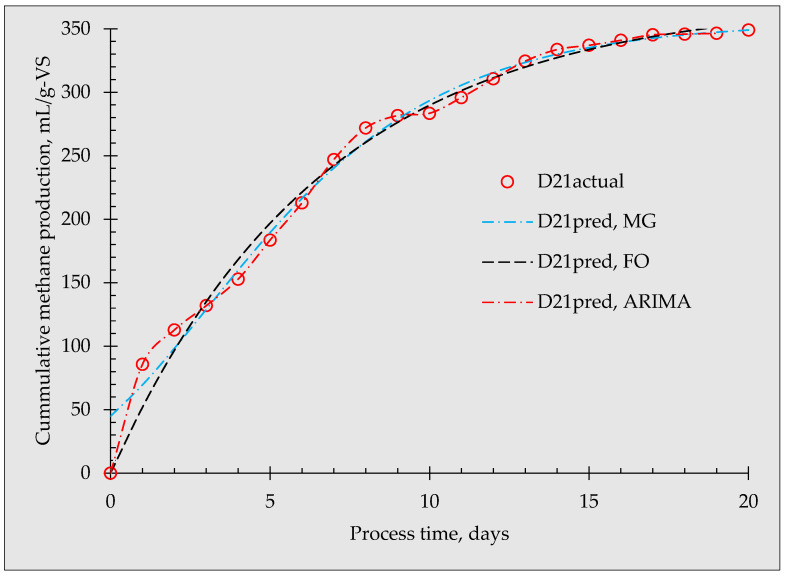
Comparative graphs of predicted biomethane production and actual biomethane production from D21 for the modified Gompertz, first-order, and ARIMA models.

**Figure 4 materials-16-06655-f004:**
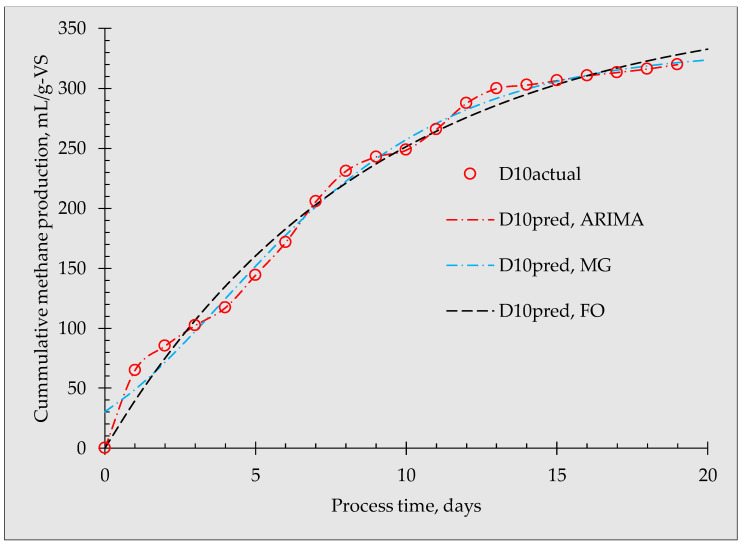
Comparative graphs of predicted biomethane production with actual biomethane production from D10 for the modified Gompertz, first-order, and ARIMA models.

**Figure 5 materials-16-06655-f005:**
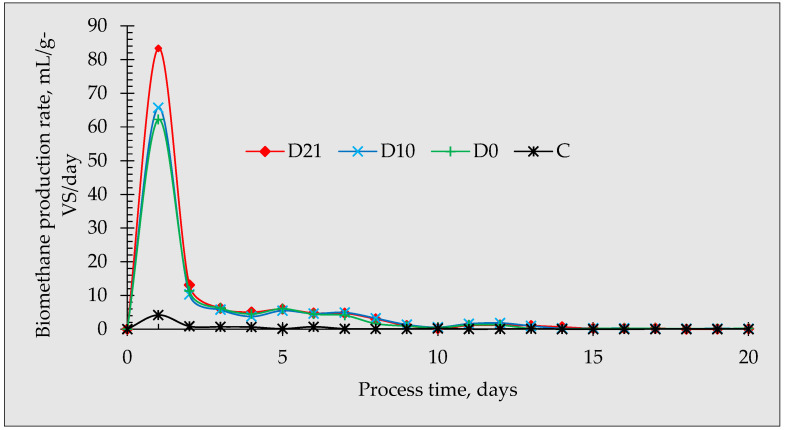
The biomethane production rate (mL/g-VS/day) of the anaerobic fermentation of glucose under different biochar incubation periods.

**Figure 6 materials-16-06655-f006:**
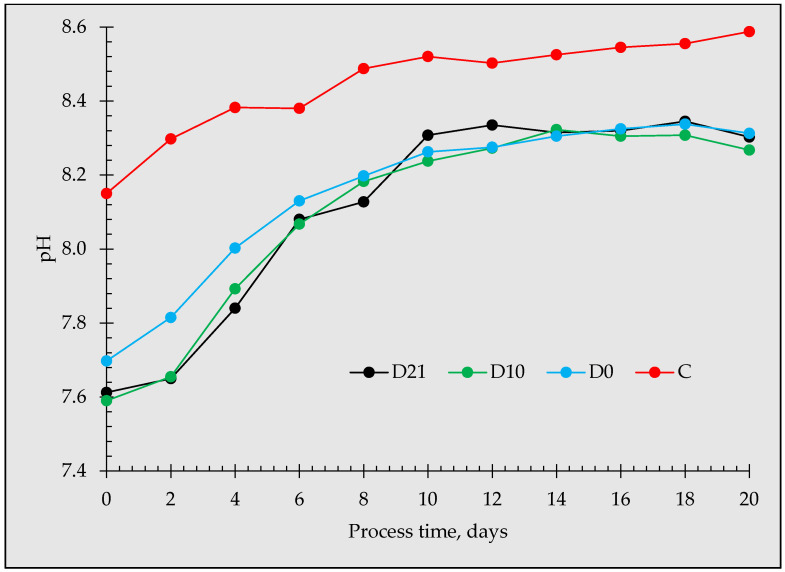
Effect of biochar incubation duration in the inoculum on the pH changes.

**Table 1 materials-16-06655-t001:** Kinetics parameters of the modified Gompertz model of the predicted biomethane production of glucose under different biochar incubation durations.

Treatment	Cumulative CH_4_ Yield		P	$R_{max}$	λ	RMSE	$R^{2}$	k
	Measured	Predicted						
D21	348.98	348.98	355.20	31.13	−1.14	10.88	0.987	0.087
D10	322.66	323.61	332.68	27.31	−0.57	8.90	0.991	0.082
D0	290.05	284.33	287.40	28.32	−0.67	7.79	0.990	0.098
C	25.15	24.17	24.78	2.03	−1.06	0.59	0.992	0.082

**Table 2 materials-16-06655-t002:** Kinetics parameters of the first-order model of the predicted biomethane production of glucose under different biochar incubation durations.

Treatment	Cumulative CH_4_ Yield		$B_{o},mL$	$k,day^{-1}$	RMSE	$R^{2}$
	Measured	Predicted				
D21	348.98	354.39	372.96	0.149	11.06	0.9861
D10	322.66	332.61	371.40	0.113	10.69	0.9867
D0	290.05	291.34	305.21	0.154	8.231	0.9890
C	25.15	24.66	26.64	0.130	0.560	0.9928

**Table 3 materials-16-06655-t003:** Parameters of the ARIMA model of the predicted biomethane production of glucose under different biochar incubation durations.

Treatment		RMSE		$R^{2}$	
	(*p*, *d*, *q*)	Validation	Test	Validation	Test
D21	5, 1, 6	0.02326	0.02256	0.99984	0.99972
D10	6, 1, 5	0.02538	0.02260	0.99984	0.99995
D0	5, 1, 1	0.02096	0.02069	0.99988	0.99995
C	5, 1, 7	0.00689	0.005812	0.99862	0.99995

## Data Availability

The data presented in this study are contained within the article.

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
