# Peer review of "Influence of Pre-Incubation of Inoculum with Biochar on Anaerobic Digestion Performance"

_materials, 2023, doi:10.3390/ma16206655_

Round 1
Reviewer 1 Report
The paper places more emphasis on the generation of models than on its intended objective “to investigate the incubation of biochar in the digestate before its application to the anaerobic digestion of glucose”. Therefore, it is recommended to conduct a title review or paper content and adjust it to align with the intended focus.
The document employs the acronym "BC," but the paper does not provide a definition for it. Therefore, it is recommended to include the definition the first time it is mentioned.
Line 59 employs the acronym "DIET," it is recommended to include the definition the first time it is mentioned.
Review the wording of lines 64 and 65; there may be a missing comma or an incomplete sentence.
Line 67 indicates that there are several reports on the use of biochar; however, only one citation is referenced. It is recommended to incorporate additional citations or, alternatively, rephrase the wording of that line.
Lines 87-103 and Figure 1 are disconnected from the article. While they may be important to justify the study's rationale, their placement at the end of the introduction severs their disconnection with the article.
Lines 105-111 consist of journal instructions and are not part of the document; they should be removed.
Lines 145-146, it is suggested that the preparation of the control be included. Why the use of a negative control was no considered?
In the results section, the authors highlight the data obtained but fail to provide a proper discussion, as there is a lack of comparison with results achieved by other authors. Additionally, there is a need to define the influence of the DIETs present in the culture medium, glucose, and the inoculum used, which is the primary focus of the article. This is because the article tends to deviate towards an extensive discussion of statistical analysis rather than addressing the initial objective.
The authors include numerous bibliographic citations, but they are not utilized in the results section.
The paper requires a more thorough review of its writing and spelling, since the paragraphs and assertions are disconnected. It´s hard to follow.
Author Response
Responses to the reviewer's comments are in the attached file.

Reviewer 2 Report
Please, revise the language since some grammar mistakes are present (e.g., line 69 “there are similar study that refers …”).
Another positive effect of BD (and carbon-based materials in general) addition to AD is the selective adsorption of toxic inhibitors (as showed in your keywords network analysis, Figure 1). Here some papers if you want to explore and add this feature:
· The role of biochar on alleviating ammonia toxicity in anaerobic digestion of nitrogen-rich wastes: A review https://doi.org/10.1016/j.biortech.2022.126924
· Semi-Continuous Anaerobic Digestion of Orange Peel Waste: Effect of Activated Carbon Addition and Alkaline Pretreatment on the Process https://doi.org/10.3390/su11123386
Please, eliminate lines 105-111.
Please, clarify in Materials and methods section how you collected the produced biogas (if it happened automatically, please, make it more clear) and how you evaluated the methane volume.
How many batch replicates did you consider in your BMP test? Please, specify in Materials and methods section.
It seems that you did not consider SMP in standard temperature and pressure (STP) conditions. If so, please, correct so that making your results more widely comparable with previous literature.
Please, mention digestate analysis of pH in Materials and methods section.
Results section is well organized but repetitive in some parts. More importantly, there is a lack of results discussion. Authors did not explain the different behaviours in terms of methane production neither with their own analysis (of microbial community, for instance, as presented in the Introduction section) or through previous literature citation. They did not make any critical comparison of their results with previous similar experiments (even solely BC addition to AD process). Moreover, there are no future perspectives or considerations (even just qualitative) about the applicability of the tested solution (what are the limitations of such solution, such as the effect of a more realistic substrate like organic waste and/or residues? what are the next experimental steps, such as semi-continuous tests?).
Please, revise the language since some grammar mistakes are present (e.g., line 69 “there are similar study that refers …”).
Author Response

(The authors gave the same response as above.)

Reviewer 3 Report
This manuscript describes the influence of the incubation of the inoculum adsorbed to biochar for a pre-acclimation period before being used in the anaerobic fermentation of glucose for methane production.
General comments:
1. As very well mentioned by the authors, AD is a process carried out by a consortium of microorganisms necessary to carry out hydrolysis, acidogenesis, acetogenesis, and methanogenesis.
In this study, presented by the authors, the first step of hydrolysis was eliminated by starting the process with glucose alone. There was no digestion of glucose. There was an anaerobic fermentation of glucose to organic acids (acidogenesis), probably followed by acetogenesis and methanogenesis to produce biogas containing methane. I perceived this work as a controlled study on the behavior of the inoculum supported on biochar, without interference from more complex and recalcitrant substrates used in anaerobic digestion, and in this way, the hydrolysis step was eliminated.
Therefore, I suggest changing the title to "Influence of the Inoculum Pre-Incubation using Biochar on the Anaerobic Digestion Performance".
2. Throughout the text, I suggest replacing "digestion time" or "digestion of glucose" with "process time" or "fermentation of glucose".
3. The suspended liquid obtained after filtration of the digestate was used as inoculum. It stopped being the digestate. Throughout the text, "digestate" should be replaced by "inoculum".
4. The biochar serves as support or adsorption for the inoculum. The incubation was of the inoculum in the biochar.
So, from everything that has been said, for example, in section "3.1 Effects of BC on the Methane Production", line 180, I suggest changing to "Figure 2 shows the cumulative methane production from the anaerobic fermentation of glucose as affected by the different periods of incubation of the inoculum in biochar.” This should be corrected throughout the text.
Specific comments:
Line 38: “The use of biochar as a supplement to enhance methane production in AD of various substrates including glucose has been established in several studies” [8], [9]. In these references, the substrates used were livestock, poultry manure and cow manure. These authors do not refer to glucose used as a substrate. I suggest replacing the references, or changing the phrase to "…of various substrates has been established.."
Line 42: Please specify “BC” the first time is mentioned in the text.
Line 105-111: Please eliminate. This is from a template.
Figure 2. Please confirm units in “mL/gVS” for Methane Production. Is this biogas or methane? Please verify carefully all the tables and text.
Line 191: Please confirm units in “mL” for cumulative biogas production. Is this biogas or methane?
Line 197: Is this biogas or methane production? Please verify carefully all the tables and text.
Table 1: Is this biogas or methane production? Please verify carefully all the tables and text.
Line 217: 372.96 units? Is this biogas or methane?
Line 277: Please verify the statement “The first order.”
Section 3.2: This section refers to the fraction of methane in biogas in %?
Line 285: Please consider “The daily fraction of produced methane during the anaerobic fermentation of glucose under various conditions is shown in Figure 6.”
Line 287: Will it be “the highest with 83.33 % followed by D10 with 65.71 %.”
Line 288: “fraction” instead of “production rate”?
Figure 6. Units of “Volume of methane per day, %” instead of "Methane Production Rate, mL/gVS.day
Line 295: Please correct “The pH variation was affected by the different biochar incubation period as shown in Figure 7.”
Lines 306-309: Please eliminate. This is methodology.
References: All references should be formatted according to journal guidelines.
Minor editing of English language required
Author Response
Please, check the attached file.

Round 2
Reviewer 1 Report
No comments.
Author Response
Please, check the attached file.

Reviewer 2 Report
In line 330 of revised manuscript it is written that "Statistical analysis demonstrated that biomethane productions were statistically different (P <0.05) among the treatments.", namely among 348.98 mL (D21), 322.66 mL (D10) and 290.05 mL (D0), if I understand correctly. However, please, specifiy somewhere in the text if the aforementioned evaluation is (most probably) limited by the fact that no batch replicates of the treatments were arranged.
Author Response
Please, check the attached file.
